# A Degradable Difunctional Initiator for ATRP That Responds to Hydrogen Peroxide

**DOI:** 10.3390/polym14091733

**Published:** 2022-04-24

**Authors:** Lawrence Hill, Hunter Sims, Ngoc Nguyen, Christopher Collins, Jeffery Palmer, Fiona Wasson

**Affiliations:** 1Department of Chemistry, Western Kentucky University, Bowling Green, KY 42101, USA; sims75@purdue.edu (H.S.); ngocng@stanford.edu (N.N.); chrischasecollins@gmail.com (C.C.); jeffery.palmer672@topper.wku.edu (J.P.); wassonfj@mail.uc.edu (F.W.); 2Department of Chemistry, Purdue University, West Lafayette, IN 47907, USA; 3Department of Chemistry, University of Cincinnati, Cincinnati, OH 45221, USA

**Keywords:** boronic ester, degradable polymer, difunctional initiator, ATRP

## Abstract

Mid-chain degradable polymers can be prepared by atom transfer radical polymerization from difunctional initiators that include triggers for the desired stimuli. While many difunctional initiators can respond to reducing conditions, procedures to prepare difunctional initiators that respond to oxidizing conditions are significantly less available in the literature. Here, a difunctional initiator incorporating an oxidizable boronic ester trigger was synthesized over four steps using simple and scalable procedures. Methyl methacrylate was polymerized by atom transfer radical polymerization using this initiator, and the polymerization kinetics were consistent with a controlled polymerization. The polymer synthesized using the difunctional initiator was found to decrease in molecular weight by 58% in the presence of hydrogen peroxide, while a control experiment using poly(methyl methacrylate) without a degradable linkage showed a much smaller decrease in molecular weight of only 9%. These observed molecular weight decreases were consistent with cleavage of the difunctional initiator via a quinone methide shift and hydrolysis of the methyl ester pendent groups in both polymers, and both polymers increased in polydispersity after oxidative degradation.

## 1. Introduction

Controlled radical polymerization techniques have allowed the design and preparation of increasingly intricate polymer architectures [1,2,3,4,5,6,7,8,9,10,11,12,13]. One major challenge is that controlled radical polymerization of vinyl monomers gives rise to polymer chains consisting of carbon–carbon bonds which are challenging to break apart, and making these polymers degradable has potential biomedical applications that could benefit from the structural diversity achievable with controlled radical polymerization. A large body of research has been devoted to this question, but one simple approach is to include a cleavable difunctional initiator to allow for the formation of vinyl polymer chains with a central degradable functionality [14,15]. This approach provides a balance between the structural control afforded by controlled radical polymerization of vinyl monomers while still resulting in a molecular weight decrease of approximately 50% in the presence of the appropriate stimuli, which was previously shown to be sufficient to trigger the release of cargo from polymer nanoparticles for drug delivery [14].

Boronic esters have also received much attention for biomedical applications due to their biocompatibility, rapid response rates, and long-term stability when stored [16,17], and boronic acid pinacol esters were recently used to synthesize block copolymers by controlled Suzuki–Miyaura cross-coupling polymerization combined with atom transfer radical polymerization (ATRP) [18]. However, to our knowledge, there is no example in the literature of a degradable difunctional initiator for controlled radical polymerization that uses a boronic ester trigger. Given the broad range of multi-stimuli responsive polymers that can be prepared from difunctional initiators using controlled radical approaches, a difunctional initiator with a boronic ester trigger may find use in a number of biomedical applications [19,20].

Here, we demonstrate the synthesis of a degradable difunctional initiator (named DFI here) for ATRP that is responsive to hydrogen peroxide by including a boronic ester trigger. We targeted a boronic ester based on the notable lack of a difunctional initiator for ATRP with this functional group in the literature despite the extensive work that was carried out on boronic ester-containing polymers for drug delivery. We began our investigation based on the report by Almutairi et al., where they synthesized degradable polyester nanoparticles via polycondensation using diol monomers with boronic ester triggers [21]. Simply adding one step to a slightly modified version of their monomer synthesis allowed for the synthesis of the DFI (Figure 1, compound 4).

This DFI incorporating an oxidizable boronic ester trigger was synthesized over four steps using simple procedures. Methyl methacrylate was then polymerized by ATRP using this initiator, and the polymerization kinetics were consistent with a controlled polymerization. Polymer synthesized using the DFI was found to decrease in molecular weight by 58% in the presence of hydrogen peroxide when dissolved in dimethylformamide, while a control experiment using poly(methyl methacrylate) without a degradable linkage showed a decrease in molecular weight of only 9%. These observed molecular weight decreases were consistent with cleavage of the DFI via a quinone methide shift and hydrolysis of the methyl ester pendent groups, and both polymers increased in polydispersity after oxidative degradation.

## 2. Materials and Methods

^1^H-NMR spectra were obtained using a 500 MHz nuclear magnetic resonance spectrometer (JEOL, Peabody, MA, USA), and ^13^C-NMR spectra were obtained using a 400 MHz nuclear magnetic resonance spectrometer (JEOL, Peabody, MA, USA). Relative polymer molecular weights were determined using an Acquity Advanced Polymer Chromatography system (size-exclusion chromatography, SEC) purchased from Waters Corporation (Milford, MA, USA) versus poly(methyl methacrylate) standards using tetrahydrofuran as the mobile phase.

Imidazole (99%), anisole (99%), and methyl methacrylate (MMA) (98%) were purchased from Acros Organics through Fisher Scientific (Pittsburgh, PA, USA). Methyl methacrylate was passed through a column of basic alumina to remove the polymerization inhibitor immediately before use. Ethyl α-bromoisobutyrate (ebib) (98%) and p-toluenesulfonic acid monohydrate (98%) were purchased from Alfa Aesar through Fisher Scientific (Pittsburgh, PA, USA). 4-bromomethylphenylboronic acid pinacol ester (97%) was purchased from AmBeed (Arlington Heights, IL, USA). Copper(I) chloride (98%), *tert*-butyldimethylsilyl chloride (TBS-Cl) (99%), and tetrahydrofuran (THF) (99.0%, HPLC grade) were purchased from Oakwood Chemical (Estill, SC, USA). 2,6-bis(hydroxymethyl)-p-cresol (95%), anhydrous dichloromethane (DCM) (99.8%), anhydrous dimethylformamide (DMF) (99.8%), anhydrous methanol (99.8%), α-bromoisobutyryl bromide (98%), N,N,N′,N″,N″-pentamethyldiethylenetriamine (PMDETA) (99%), and hydrogen peroxide (30 wt.%) were purchased from Sigma Aldrich (Milwaukee, WI, USA). Copper (I) bromide (98%) was purchased from Strem Chemicals (Newburyport, MA, USA). Hexane, anhydrous sodium sulfate, glacial acetic acid, basic alumina, and anhydrous potassium carbonate were standard reagent grade, and anhydrous potassium carbonate was ground in a mortar and pestle and dried for 2.5 h at 100 °C before use. All liquids were handled in a ventilated fume hood. Copper(I) chloride and copper(I) bromide were stirred in glacial acetic acid overnight, washed with ethanol, filtered, dried under vacuum, and stored under argon before use.

**Compound 1.** Compound 1 was synthesized based on Almutairi et al [21]. 2,6-bis(hydroxymethyl)-p-cresol (2.5130 g, 14.94 mmol) and imidazole (2.270 g, 32.71 mmol) were dissolved in anhydrous DMF (7.5 mL) in a 100 mL round bottom flask with stir bar and chilled in an ice bath for 10 min resulting in a yellow solution with black flecks. A solution of TBS-Cl (4.9416 g, 32.79 mmol) dissolved in anhydrous DMF (12.5 mL) was added dropwise via syringe to the cooled, stirring, solution for approximately 8 min, and then stirred in the ice bath for an additional 10 min. The flask was then removed from the ice bath and placed in a sonication bath for 10 min to break up the precipitate. This mixture was then stirred at room temperature for an additional 2 h. The cloudy yellow mixture with white solids was filtered and the white solids were rinsed with hexane (40 mL) into a separatory funnel. The two liquid phases were mixed by shaking and the hexane phase was collected. The remaining DMF phase was washed with additional hexane (20 mL, 2×), the hexane phases were combined, and the hexane solution was placed on a rotary evaporator at 45 °C until approximately 8 mL of bright yellow solution remained. This solution was washed with saturated potassium carbonate (3 × 10 mL, 0.25 M) and dried over sodium sulfate for 20 min before removing the hexane under reduced pressure (45 °C) to give a yellow oil. This oil was dried under vacuum at room temperature overnight to give compound 1 as a slightly yellow oil (5.2334 g, 88%).

**Compound 2.** Compound 2 was synthesized based on Almutairi et al [21]. Anhydrous DMF (15 mL) was combined with anhydrous potassium carbonate (1.1760 g, 8.51 mmol) in a 100 mL round bottom flask, sealed with septum, and placed in a sonication bath at 50 °C for 1.5 h resulting in a cloudy yellow mixture with visible white solids. The round bottom flask was cooled to 0 °C in an ice bath and compound 1 (2.8167 g, 7.10 mmol) was added quickly by syringe before an additional 2 mL anhydrous DMF was used to wash the syringe. The mixture was stirred for 10 min at 0 °C before 4-bromomethylphenylboronic acid pinacol ester (2.2140 g, 7.46 mmol) was added. The mixture was stirred for another 10 min at 0 °C, allowed to warm to room temperature, and left to stir overnight (18 h) at room temperature. The crude reaction mixture was filtered, and the solids were rinsed with hexane (70 mL) into a separatory funnel. The two liquid phases were mixed by shaking and the hexane phase was collected. The remaining DMF phase was washed with additional hexane (20 mL, 1×). The hexane phases were combined, washed into a separatory funnel with 5 mL hexane, and washed with brine (20 mL, 2×, 29 g NaCl/100 mL H_2_O). The hexane solution was dried over sodium sulfate for 45 min and the solvent evaporated under reduced pressure at 45 °C. The resulting oil was dried under vacuum overnight at 50 °C to give a partially solidified, cloudy oil (3.8875 g, 89%).

**Compound 3.** Compound 3 was synthesized based on Almutairi et al [21]. Compound 2 (1.5024 g, 2.45 mmol) was dissolved in anhydrous methanol (8 mL) in a 20 mL glass vial with stir bar, and p-toluenesulfonic acid monohydrate (0.094 g, 0.49 mmol) was added. The vial was placed in a sonication bath for 20 min and then stirred at room temperature for 2 h. The solvent was removed under vacuum (50 °C) and the resulting oil was dissolved in 2 mL anhydrous methanol and purified using column chromatography (silica, hexane/ethyl acetate = 1:1) to give a cloudy oil that was dried under vacuum at 50 °C for 4 h to give compound 3 (0.9302 g, 99%).

We also had success purifying compound 3 by liquid/liquid extractions with methanol and hexane instead of column chromatography. In these cases, the methanol phase was collected and evaporated under reduced pressure. The resulting oil was then dissolved in DCM, washed with 0.25 M potassium carbonate solution, dried over sodium sulfate, and the solvent evaporated under reduced pressure at 50 °C to give a yellow oil. This procedure gave NMR spectra with some small amounts of impurities that did not interfere with the synthesis of compound 4.

**Compound 4 (DFI).** Compound 3 (0.6251 g, 1.63 mmol) was combined with triethylamine (0.54 mL, 4.24 mmol) in a 250 mL round bottom flask with stir bar and anhydrous dichloromethane (20 mL) was added. The flask was cooled in an ice bath for 10 min before a solution of α-bromoisobutyryl bromide (0.48 mL, 3.41 mmol) dissolved in anhydrous dichloromethane (5 mL) was added and dropwise over 10 min. The solution was stirred for another 10 min in the ice bath and then left to stir at room temperature overnight. The crude product was washed into a separatory funnel with 5 mL dichloromethane and then washed with water (30 mL, 2×). The organic phase was collected and evaporated under reduced pressure to obtain a dark yellow oil. This oil was combined with 2 mL methanol and placed in a sonication bath for 30 min to give a white precipitate. The solids were collected by vacuum filtration and washed with an additional 2 mL methanol before drying under vacuum at room temperature overnight to give compound 4 (0.7147 g, 65%). ^1^H-NMR (500 MHz, Chloroform-D) δ 7.84 (d, *J* = 7.6 Hz, 2H), 7.46 (d, *J* = 8.5 Hz, 2H), 7.25 (s, 2H), 5.24 (s, 4H), 5.01 (s, 2H), 2.34 (s, 3H), 1.93 (s, 12H), 1.35 (s, 12H). ^13^C-NMR (101 MHz, Chloroform-D) δ 171.52, 153.94, 139.94, 135.17, 134.43, 131.71, 128.84, 127.01, 83.95, 77.31, 63.06, 55.74, 30.87, 24.97, 21.00.

The procedure above represents our best yield obtained for the synthesis of compound 4, but we found that the precipitation from methanol was unsuccessful for some reactions. The reason for this is unknown at this time, but samples that failed to precipitate from methanol were instead evaporated, dissolved in dichloromethane, and purified by column chromatography (silica, hexane/ethyl acetate = 4:1) which resulted in lower yields (15–27%) than precipitation from methanol.

**pMMA-DFI-pMMA.** A stock solution of the ligand N,N,N′,N″,N″-pentamethyldiethylenetriamine (PMDETA) was prepared in anisole (0.38 M). All solutions and neat liquids were sparged with argon for 20 min prior to use. 

In a typical reaction, copper (I) chloride (29.3 mg, 0.296 mmol) and the DFI (compound 4, 84.5 mg, 0.124 mmol) were added to a 100 mL Schlenk flask with stir bar, the flask was sealed with a septum secured by parafilm, and the air was removed by vacuum and replaced with argon for a total of five cycles. The PMDETA/anisole stock solution (0.65 mL, 0.25 mmol PDMETA) was injected under argon with gentle stirring to form a green/yellow solution, followed by an additional 5 mL deoxygenated anisole. Methyl methacrylate (5.3 mL, 49.8 mmol) was then injected under argon, followed quickly by an additional 5 mL deoxygenated anisole added along the interior sides of the flask to rinse the contents to the bottom of the flask. The flask was then lowered into a heated oil bath and maintained at approximately 55–65 °C with vigorous stirring. Small aliquots were periodically removed under argon to monitor the reaction progress for initial kinetic studies. These samples were passed through a small plug of basic alumina followed by a 0.45 µm PTFE syringe filter before characterization by ^1^H-NMR and size exclusion chromatography. These kinetic studies allowed us to correlate molecular weight with reaction time. Subsequent polymerization reactions were then conducted without removing samples, and these reactions were quenched at a predetermined time by cooling to room temperature, removing the septum, and diluting with dichloromethane.

The contents of the Schlenk flask were then poured through basic alumina in a fritted funnel with additional dichloromethane, and the filtrate was transferred to a round-bottom flask before removing the solvent under reduced pressure to near dryness. The visible solids were dissolved in a minimal amount of dichloromethane and added dropwise to a large excess of vigorously stirred methanol. The poly(methyl methacrylate) precipitated as white strands in the beaker that were collected by vacuum filtration and dried under vacuum overnight to give a white solid.

**ebib-pMMA.** Stock solutions of the ligand N,N,N′,N″,N″-pentamethyldiethylenetriamine (PMDETA) and the initiator ethyl α-bromoisobutyrate (ebib) were prepared separately in anisole at concentrations of 0.38 M. All solutions and neat liquids were sparged with argon for 20 min prior to use.

In a typical reaction, copper(I) chloride (39 mg, 0.39 mmol) was added to a 100 mL Schlenk flask with stir bar, the flask was sealed with a septum and parafilm, and the air was removed by vacuum and replaced with argon for a total of five cycles. The PMDETA stock solution (0.75 mL, 0.28 mmol PDMETA) was injected under argon with gentle stirring to form a green solution, followed by adding anisole (12 mL) and then methyl methacrylate (6.0 mL, 56 mmol). The ebib stock solution (0.75 mL, 0.28 mmol ebib) was injected last under argon. The flask was then lowered into a heated oil bath and maintained at approximately 55–65 °C with vigorous stirring. Small aliquots were periodically removed under argon to monitor the reaction progress for initial kinetic studies. These samples were passed through a small plug of basic alumina followed by a 0.45 µm PTFE syringe filter before characterization by ^1^H-NMR and size exclusion chromatography. These kinetic studies allowed us to correlate molecular weight with reaction time. Subsequent polymerization reactions were then conducted without removing samples, and these reactions were quenched at a predetermined time by cooling to room temperature, removing the septum, and diluting with dichloromethane.

The contents of the Schlenk flask were then poured through basic alumina in a fritted funnel with additional dichloromethane, and the filtrate was transferred to a round-bottom flask before removing the solvent under reduced pressure to near dryness. The visible solids were dissolved in a minimal amount of dichloromethane and added dropwise to a large excess of vigorously stirred methanol. The poly(methyl methacrylate) precipitated as white strands in the beaker that were collected by vacuum filtration and dried under vacuum overnight to give a white solid.

Methyl methacrylate was also polymerized from ethyl α-bromoisobutyrate using copper(I) bromide, which resulted in a less controlled polymerization than using copper(I) chloride [22,23]. However, changing the halide did not affect the outcome of polymer degradation experiments of ebib-pMMA.

**Polymer degradation experiments.** Conditions for the polymer degradation experiments were adapted from Almutairi et al [21]. Approximately 10 mg of purified polymer was weighed into a vial, and for each milligram of polymer, 1 mL of N,N-dimethylformamide (DMF), and 10 μL of 30 wt.% hydrogen peroxide were added, giving a peroxide concentration of approximately 100 mM. In order to allow the polymer to fully degrade, the solution was stirred for four days before the DMF and leftover hydrogen peroxide were removed under vacuum on a rotary evaporator with a heat gun in a chemical fume hood. The resulting residue was dissolved in tetrahydrofuran for molecular weight characterization.

## 3. Results and Discussion

### 3.1. Synthesis of the DFI

The DFI was synthesized over four steps (Figure 1). Compounds 1–3 were synthesized based on the procedure given by Almutairi et al [21]. In the first step, the benzyl alcohol groups of 2,6-bis(hydroxymethyl)-p-cresol were protected with TBS-Cl using imidazole in DMF to form compound 1 in good yields. Compound 1 was then coupled to 4-bromomethylphenylboronic acid pinacol ester by reaction of the phenolic oxygen with the benzyl bromide in DMF with dry potassium carbonate to afford compound 2 in good yields. The TBS protecting groups were then removed from compound 2 using TsOH in methanol to afford compound 3 in nearly quantitative yields. Esterification of the benzylic alcohols of compound 3 with α-bromoisobutyryl bromide in DCM with triethylamine afforded the DFI compound 4.

The synthesis of the DFI was straightforward and all the reactions were conducted by undergraduate and high school student researchers. The most reproducible procedures that resulted in our best yields are given in the Materials and Methods section, though these yields varied slightly. We found the synthesis of compounds 1–3 to be easily scaled up without other changes to the procedure. While we initially used a column to purify compound 3, we found that the need for column chromatography could be reassessed after the reaction to form the DFI (compound 4), which facilitated increasing the scale of the reactions. The reaction to form compound 4 was also scalable, and we had some success recrystallizing this product to make the entire synthesis free of chromatography at scale. However, for reasons still unknown to us, some reactions to form compound 4 did not readily form a precipitate of the product and required column chromatography, which limited us to approximately 1 g per purified batch in these cases. The simplicity of the synthesis should allow others to easily prepare this initiator for their own use.

The ^1^H-NMR spectrum of the DFI with corresponding spectral assignments is shown in Figure 2, and the ^13^C-NMR spectrum of the DFI with corresponding spectral assignments is shown in Figure 3. Integration values and chemical shifts of the ^1^H-NMR spectrum matched expected values for the DFI with no discrepancies. Similarly, the chemical shifts observed in the ^13^C-NMR spectrum were all consistent with the expected structure. Additionally, the ^13^C-NMR spectrum, shown in Figure 3, was obtained from a sample of the DFI that had been stored in a vial in a dark cabinet for approximately 2 years, which indicates the stability of this initiator when stored as a solid.

### 3.2. Polymerization Using the DFI

Methyl methacrylate was polymerized using the DFI with copper(I) chloride, PMDETA, and anisole (Figure 4a). Copper chloride was used to obtain a polymer with lower polydispersity via halogen exchange [22,23]. We did not conduct end group analysis on these materials, but we expect a preferential exchange from bromine to chlorine in the polymer based on previous studies with small molecule model systems [24]. A single, mostly symmetrical SEC trace was obtained for most samples, providing evidence that the polymerization was controlled using the DFI, though a high molecular weight shoulder began to appear at higher monomer conversion (Figure 4b). This high molecular weight shoulder was small compared to the overall molecular weight distribution, but it continually shifted to higher molecular weight with increasing monomer conversion. One possible explanation for this shoulder is that there may be a small amount of coupling between growing polymer chains. Because each polymer chain has two living ends, any coupling of two chains forms a new polymer chain with two living ends that can continue to grow as the reaction proceeds. Here, the presence of this high MW shoulder did not interfere with the experiments that were central to this work, but the polymerization conditions could be optimized further to limit this coupling if desired. Radical termination of pMMA occurs preferentially through disproportionation over chain coupling, but some amount of chain coupling is still expected to occur [25].

A linear correlation was observed between M_n_ and monomer conversion and the polydispersity decreased with conversion, indicating that the polymerization was controlled (Figure 4c). This is consistent with termination being a minor contributor to the overall polymerization process. The polymerization kinetics were first-order, which also indicated a constant radical concentration during the polymerization (Figure 4d). The initiator efficiency was estimated to be 79% by comparing the measured molecular weight of 24,670 g/mol at 47% conversion with the theoretical molecular weight of 19,463 based on the initial ratios of monomer to the initiator. This estimated initiator efficiency is comparable to that observed for other ATRP initiators [5]. The polymer obtained was purified by passing through basic alumina followed by precipitation from methanol to remove unreacted monomer before further use in oxidative degradation studies.

### 3.3. Oxidative Degradation of pMMA with and without the DFI

With the polymerization conditions in hand, we synthesized pMMA-DFI-pMMA and ebib-pMMA to use for oxidative degradation tests (Figure 5a,b). For these studies, polymers were stirred in DMF with approximately 100 mM H_2_O_2_ for approximately 90 h before the solvent was removed and the residue was dissolved in THF and analyzed using size exclusion chromatography. These two polymers responded very differently to this oxidation condition as expected from the inclusion of a mid-chain degradable linkage in pMMA-DFI-pMMA that was not present in ebib-pMMA. The number average molecular weight of pMMA-DFI-pMMA decreased by 58%, which is consistent with degradation through oxidative cleavage of the central linkage due to rearrangement of the boronic ester trigger in addition to hydrolysis of the methyl ester pendent groups. By comparison, only a 9% decrease in M_n_ was observed for ebib-pMMA, which is consistent with the hydrolysis of the methyl ester pendent groups without cleavage of the polymer backbone. This result demonstrated that the DFI was capable of acting as a degradable linkage under these conditions.

The mechanism for the oxidative degradation of aryl boronic esters with peroxides was extensively studied by others [17,21,26,27,28]. This mechanism begins with the insertion of oxygen into the bond between boron and the aryl carbon atom and the subsequent formation of a phenoxide ion. A quinone methide rearrangement then cleaves the benzyl ester bonds in the polymer backbone resulting in a decrease in polymer molecular weight. We did not examine the mechanism further in our work which was focused on communicating the synthesis of the DFI and demonstrating its utility for synthesizing mid-chain degradable polymers by ATRP.

## 4. Conclusions

A difunctional initiator for ATRP was synthesized and methyl methacrylate was polymerized using this initiator. Polymer synthesized using the difunctional initiator was found to decrease in molecular weight in the presence of hydrogen peroxide when dissolved in dimethylformamide. The molecular weight decrease was consistent with both the cleavage of the boronic ester via a quinone methide shift and the hydrolysis of the methyl ester pendent groups. Poly(methyl methacrylate) synthesized using a monofunctional initiator (ebib) showed a much smaller decrease in molecular weight in the presence of hydrogen peroxide when dissolved in dimethylformamide that was consistent with the hydrolysis of the methyl ester pendent groups. Both polymers increased in polydispersity after oxidative degradation. Future planned studies include determining the degradation rates of these polymers, optimizing degradation conditions to accentuate differences between degradation rates of different initiators, and incorporating antioxidant pendent groups into polymers to perturb the degradation rate of this initiator. The DFI reported here can, in principle, be used as a difunctional monomer for polyaddition reactions to incorporate many degradable linkages per polymer chain [29,30,31,32,33,34,35,36].

## Figures and Tables

**Figure 1 polymers-14-01733-f001:**
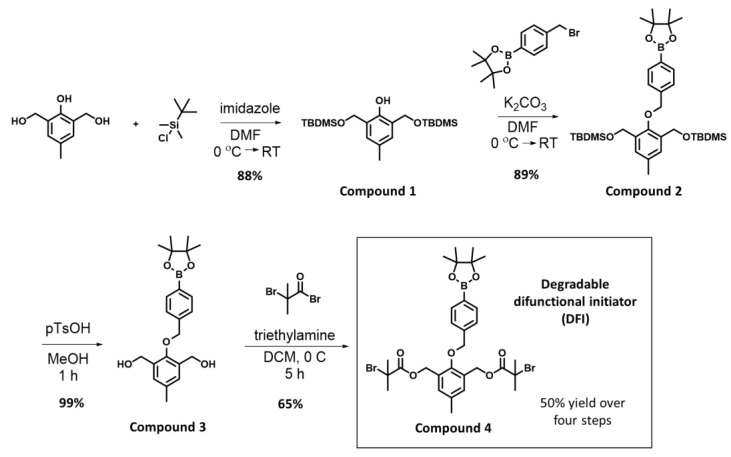
Synthesis of the degradable difunctional initiator (DFI) over four steps.

**Figure 2 polymers-14-01733-f002:**
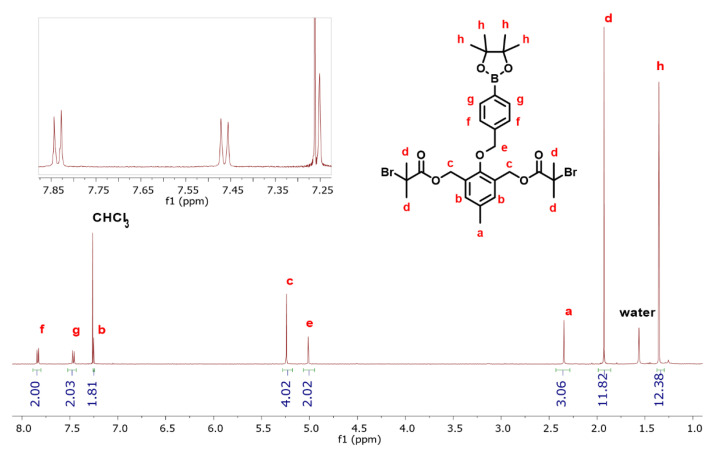
^1^H-NMR spectrum of the difunctional initiator (DFI) compound 4 in CDCl_3_.

**Figure 3 polymers-14-01733-f003:**
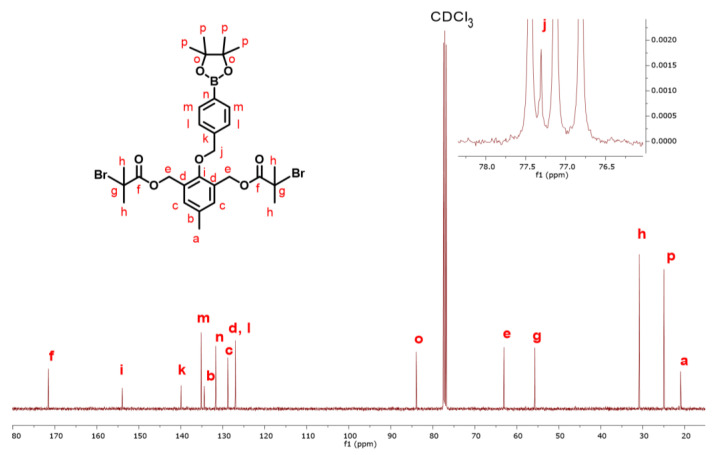
^13^C-NMR spectrum of the difunctional initiator (DFI) compound 4 in CDCl_3_.

**Figure 4 polymers-14-01733-f004:**
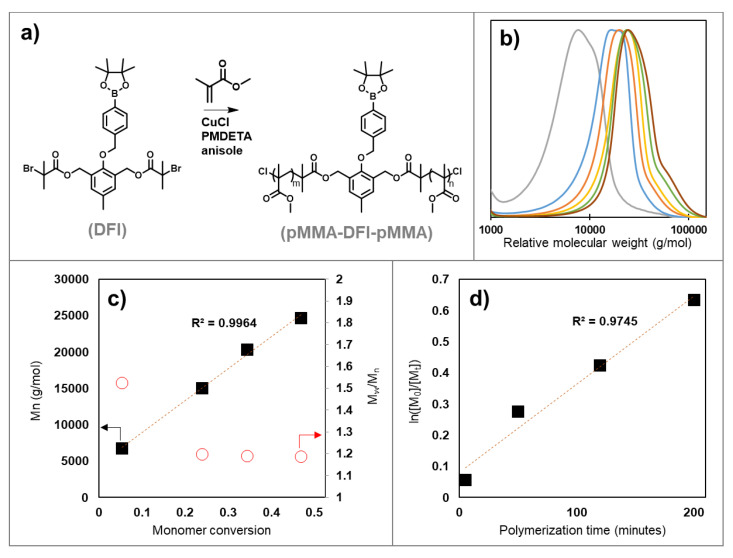
Monitoring the evolution of molecular weight and polydispersity in the ATRP of methyl methacrylate using the difunctional initiator (DFI): (**a**) Polymerization of methyl methacrylate using the DFI, T = 55 °C, [MMA]_0_/[DFI]_0_/[CuCl]_0_/[PMDETA]_0_ = 400/1/2.24/2; (**b**) SEC traces obtained from samples taken at 15, 50, 95, 120, 160, and 200 min for a representative synthesis of pMMA-DFI-pMMA (versus pMMA standards in THF); (**c**) Evolution of molecular weight (solid black squares) and polydispersity (hollow red circles) versus monomer conversion in the ATRP of MMA during a representative polymerization; (**d**) First-order kinetic plot for the ATRP of MMA using the DFI.

**Figure 5 polymers-14-01733-f005:**
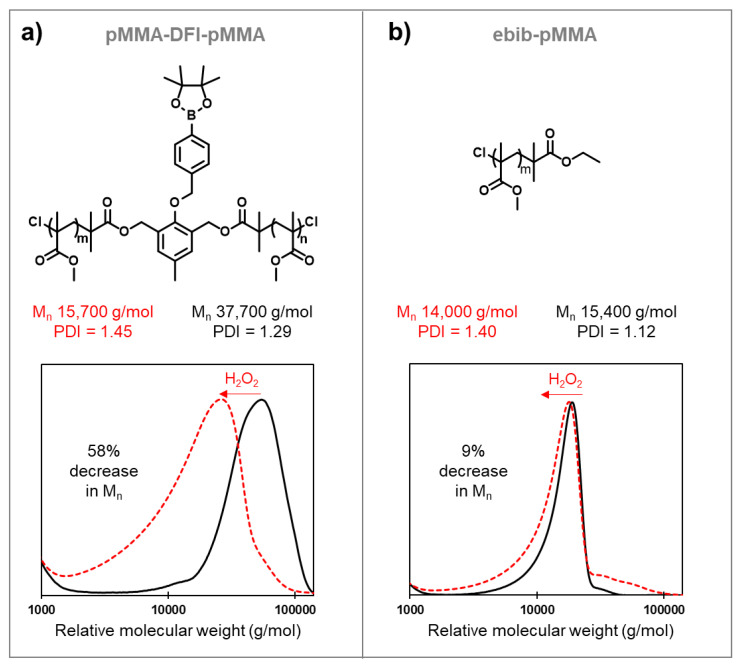
Molecular weights and polydispersity indexes for poly(methyl methacrylate) synthesized by ATRP using two different initiators before and after oxidative degradation experiments: (**a**) pMMA-DFI-pMMA; (**b**) ebib-pMMA. Plots of relative molecular weight distributions were obtained versus poly(methyl methacrylate) standards in tetrahydrofuran.

## Data Availability

Not applicable.

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
