# Peer review of "A Degradable Difunctional Initiator for ATRP That Responds to Hydrogen Peroxide"

_polymers, 2022, doi:10.3390/polym14091733_

Round 1

Reviewer 1 Report

In this manuscript, Hill et al. demonstrated the synthesis of a degradable DFI for ATRP and found the prepared polymer can be decreased in molecular weight by 58% in hydrogen peroxide. The synthetic procedure of DFI in this manuscript is simple and scalable. Besides, the preparation of degradable polymers is indeed good for environmental friendliness. So, this manuscript can be considered for the publication after addressing some concerns as follow:

  1. Since Dr. Wang first discovered ATRP in Matyjaszewski's group in 1995, great progresses have been made in this field. Particularly, both Prof. Matyjaszewski and Prof. Sawamoto have made great contributions. Their pioneering work should be referred and cited.
  2. In Figure 6, the degradation process would generate six products, four of which are small molecules. Are there some evidences to confirm these four molecules? Such as GC-MS or NMR results?
  3. In Figure 6, why did there be the number “2” in front of MeOH? As we known, the ester in the polymer can be degraded to generate carboxylic acids and a large number of MeOH.
  4. Although a few bonds were broken in the oxidative degradation, all these C-C bonds generated in the polymerization did not break. Trying to cut off these C-C bonds generated in polymerization is the key problem of degradation. The authors should not ignore this major challenge in degradation process.
  5. There are some mistakes in the manuscript, such as “Figure 3b”, “Figure 3c”. Abbreviations should be used after the first occurrence of the word. When the word reappears later, please use abbreviations, such as “DFI”, “ATRP”. Please do proofread and improve this manuscript carefully.

Reviewer 3 Report

In this manuscript, Hill et al. reported atom transfer radical polymerization of methyl methacrylate (MMA) by using a degradable difunctional initiator. The authors focused on the initiation, polymerization and mechanism of oxidative degradation. The paper’s idea is novel. However, the highlighted method for investigating characterization of the polymers and the mechanism of oxidative degradation was only SEC measurements. MALDI TOF MS and NMR measurements are required to clarify the chain-end structures of the polymers. It's also difficult to make conclusion for the mechanism of oxidative degradation, which should be further characterized by other measurements, such as MALDI TOF MS, IR, NMR and so on. Based on the SEC results, the oxidative degradation mechanism is not clear. The manuscript is not suitable for publication as a research paper in Polymers.

Round 2

Reviewer 1 Report

Now it can be published.

Reviewer 3 Report

I have no question.